# Analysis of the Association Between MicroRNA Biogenesis Gene Polymorphisms and Venous Thromboembolism in Koreans

**DOI:** 10.3390/ijms20153771

**Published:** 2019-08-01

**Authors:** Eun Ju Ko, Eo Jin Kim, Jung Oh Kim, Jung Hoon Sung, Han Sung Park, Chang Soo Ryu, Jisu Oh, So Young Chong, Doyeun Oh, Nam Keun Kim

**Affiliations:** 1Department of Biomedical Science, College of Life Science, CHA University, Seongnam 13488, Korea; 2Department on Internal Medicine, Asan Medical Center, University of Ulsan College of Medicine, Seoul 05505, Korea; 3Department of Internal Medicine, CHA Bundang Medical Center, CHA University, Seongnam 13496, Korea

**Keywords:** vascular diseases, venous thromboembolism, microRNAs, 3′ untranslated regions, polymorphism, single nucleotide

## Abstract

Venous thromboembolism (VTE) involves the formation of a blood clot, typically in the deep veins of the leg or arm (deep vein thrombosis), which then travels via the circulatory system and ultimately lodges in the lungs, resulting in pulmonary embolism. A number of microRNAs (miRNAs) are well-known regulators of thrombosis and thrombolysis, and mutations in miRNA biogenesis genes, such as *DICER1*, *DROSHA* have been implicated in miRNA synthesis and function. We investigated the genetic association between polymorphisms in four miRNA biogenesis genes, *DICER1* rs3742330A > G, *DROSHA* rs10719T > C, *RAN* rs14035C > T and *XPO5* rs11077A > C, and VTE in 503 Koreans: 300 controls and 203 patients. Genotyping was assessed with polymerase chain reaction-restriction fragment length polymorphism assays. We detected associations between polymorphisms in *RAN* and *XPO5* and VTE prevalence (*RAN* rs14035CC + CT versus TT: *p* = 0.018; *XPO5* rs11077AA + AC versus CC: *p* < 0.001). Analysis of allele combinations of all four polymorphisms (*DICER1*, *DROSHA*, *RAN,*
*XPO5*) revealed that A-T-T-A was associated with decreased VTE prevalence (*p* = 0.0002), and A-T-C-C was associated with increased VTE prevalence (*p* = 0.027). Moreover, in subjects with provoked VTE, the *DROSHA* rs10719T > C, polymorphism was associated with increased disease prevalence (TT versus TC + CC: *p* < 0.039). Our study demonstrates that *RAN* and *XPO5* polymorphisms are associated with risk for VTE in Korean subjects.

## 1. Introduction

Venous thromboembolism (VTE), which involves deep vein thrombosis (DVT) followed by pulmonary embolism (PE), is the third most common cardiovascular disease following myocardial infarction and stroke [1]. DVT is a disease that occurs when blood clots form in the deep veins, usually in the legs. The thrombus can then become dislodged and migrate to form a blockage in the pulmonary veins, with 45.4% of DVT patients ultimately developing PE [2,3]. The incidence of VTE is higher in western populations than in Asians, although recently, annual increases in the incidence of VTE have been noted in Asian individuals, reaching levels as high as that of western people under certain circumstances [4,5]. VTE can be classified into two groups, known as provoked and unprovoked [6]. The provoked group includes patients with a known cause of a hypercoagulable state, including female hormones (estrogens or oral contraception), surgery (post-operative period, particularly for orthopedic surgery of the hip or knee, or urinary surgery), pregnancy, phospholipid antibodies in the blood (e.g., anti-cardiolipin antibodies or lupus anticoagulant), cancer, high levels of blood homocysteine, or inherited protein deficiencies associated with coagulation [2,7]. In contrast, unprovoked VTE occurs in patients with no clear causality. Pro-clinical symptoms of DVT include swelling, redness, tenderness of the leg, and manifestations of PE, such as syncope, wheezing, substernal pain, hemoptysis, tachycardia, or tachypnea [8]. However, because these symptoms are not specific to VTE, only about 20% of patients with DVT or PE are properly diagnosed [2,9,10].

The underlying causes of thrombosis are referred to as Virchow’s Triad; these include endothelial injury, abnormal blood flow, and hypercoagulation [8]. Notably, a previous study found that microRNAs (miRNAs) are associated with hemostasis [11] and are involved in the formation of blood clots, the hypercoagulable state, and endothelial cell proliferation [12,13,14,15,16]. These miRNAs species are short, non-coding RNAs, 22–24 nucleotides in length, which control gene expression as post-transcriptional regulators. The miRNA-encoding genes are transcribed by RNA polymerase II as long, primary miRNAs (pri-miRNA) with a stem-loop structure (100–1000 nts), that are cleaved by the DROSHA/DGCR8 complex in nucleus. The resulting 80-nt, hairpin-shaped precursor miRNAs (pre-miRNAs) are exported to the cytoplasm by the Ras-related nuclear protein (RAN) GTPase and exportin 5 (XPO5). There, they are processed by the DICER1/TRBP complex to form miRNA duplex. One strand of the miRNA duplex is loaded onto Argonaute (AGO) to form the RNA-induced silencing complex (RISC), which binds to the 3′-untranslated region (UTR) of an mRNA transcript and functions to suppress translation or promote mRNA degradation, through a mechanism known as post-transcriptional gene silencing (PTGS) [17,18,19]. Regulation by miRNA-mediated PTGS plays a fundamental role in many essential cellular processes, and because of this, proper miRNA biogenesis is critical for maintaining metabolic balance [20,21,22].

The miRNA biogenesis genes are critical for miRNA synthesis, and mutations in these genes can have global effects on miRNA expression [23,24,25,26,27]. For example, smoking-induced damage to the gene encoding DICER1 results in altered alveolar macrophage miRNA expression [28]. Mutations in *DROSHA* are associated with effects on male fertility, and in particular, a lack of DROSHA in the male germline causes defects in miRNA production and spermatogenesis [29]. In addition, mutations in *XPO5* have been implicated in the development and progression of certain cancers. Specifically, in hepatocellular carcinoma, a mutant form of this protein is aberrantly phosphorylated by extracellular signal-regulated kinase (ERK) kinase, undergoes a conformational change, and loses the ability to transport precursor miRNAs. As a result, pre-miRNA is retained in the nucleus, and global miRNA expression levels are down-regulated [30].

A number of studies have also shown that miRNAs play important roles conditions associated with increased risk for VTE, such as diabetes mellitus, hypertension, and hypercholesterolemia [31,32,33], as well as in thrombosis development directly. One previous study, in particular, revealed that miRNA-145 is associated with thrombus formation, with lower levels present in VTE patients. This miRNA represses expression of tissue factor (TF), and it is proposed that miRNA-145-mediated down-regulation of TF is required for proper function of the intrinsic coagulation pathway and the suppression of thrombus production [14]. Several other miRNAs are also associated with various aspects of vascular biology and hemostasis, including platelet biogenesis (megakaryopoiesis) and function, expression of coagulation factor and anticoagulation factor, and fibrinolysis. For example, miRNA-23 and miRNA-27 are associated with pro-angiogenesis functions, whereas miRNA-17, miRNA-92, and miRNA-32 are involved in anti-angiogenesis. In addition, miRNA-223 plays an important role in platelet formation and thrombin activation, and both miRNA-19b and miRNA-20a are associated with pro-coagulant activity in anti-phospholipid syndrome (APS) and systemic lupus erythematous (SLE) patients [15,16].

In our previous study, we detected an association between 3′-UTR polymorphisms in miRNA biogenesis genes and ischemic stroke risk and prognosis. In particular, polymorphisms at *DICER1* rs3742330 and *DROSHA* rs10719 were found to be associated with ischemic stroke susceptibility, and the *RAN* rs14035 polymorphic locus was significantly linked to post-stroke mortality [34]. Based on these findings, in this study, we selected four genes (*DICER1*, *DROSHA*, *RAN*, *XPO5*) among several miRNA biogenesis genes. The *DROSHA* gene is located on 15p13.3 and DROSHA is a nuclear ribonulcease III (RNase III) enzyme, which cleaves a stem-loop pri-miRNAs to hairpin-shaped pre-miRNAs. *DICER1* gene is located on 14q32.13 and DICER1 is another RNase III, which cleaves the pre-miRNA to intermediate miRNA duplex [35]. R*AN* gene is located on 12q24.33 and RAN is a GTPase that makes a pre-miRNA/XPO5/Ran-GTP ternary complex and transports pre-miRNA to cytoplasm. *XPO5* gene is located on 6p21.1 and XPO5 is a member of the karyopherin family and exports pre-miRNAs from the nucleus to cytoplasm [36]. Moreover, we investigated 3′-UTR polymorphisms of this four miRNA biogenesis gene, including *DICER1* rs3742330A > G, *DROSHA* rs10719T > C, *RAN* rs14035C > T, and *XPO5* rs11077A > C, which are independently or complementarily associated with disease prevalence and clinical outcomes in VTE patients.

## 2. Results

### 2.1. Characteristics of the Study Population.

The demographic characteristics for VTE patients and controls are listed in Table 1. There were no significant differences in age, hypertension, hyperlipidemia, or smoking status between the two groups. In contrast, VTE was more prevalent among men, and the patient group included significantly more subjects with diabetes.

### 2.2. Genotype Frequencies of miRNA Biogenesis Genes

We investigated the prevalence of the *DICER1* rs3742330A > G, *DROSHA* rs10719T > C, *RAN* rs14035C > T, and *XPO5* rs11077A > C gene polymorphisms in VTE patients and control subjects. Table 2 shows the genotype distributions in each of these groups. We found that the *XPO5* rs11077A > C polymorphism was associated with an increased risk for VTE (AA versus AC: adjusted odds ratio (AOR) = 2.522, 95% confidence interval (CI) = 1.564–4.067, *p* < 0.001; AA versus AC + CC: AOR = 2.493, 95% CI = 1.552–4.003, *p* < 0.001). In contrast, the *RAN* CT genotype was more frequent in control group than VTE patients, and the *RAN* rs14035C > T polymorphism was associated with reduced VTE risk (CC versus CT: AOR = 0.630, 95% CI = 0.425–0.935, *p* = 0.022; CC versus CT + TT: AOR = 0.627, 95% CI = 0.427–0.922, *p* = 0.018). The *XPO5* rs11077A > C polymorphism showed the same increased AOR pattern in subjects with both provoked and unprovoked VTE, whereas the *RAN* rs14035C > T polymorphism also demonstrated a reduced AOR in subjects with unprovoked VTE. The *DICER1* rs3742330A > G and *DROSHA* rs10719T > C polymorphisms were not significantly correlated with either total or unprovoked VTE occurrence. However, for provoked VTE, the *DROSHA* rs10719T > C polymorphism was associated with an increased disease risk (TT versus TC + CC: AOR = 2.460, 95% CI = 1.048–5.774, *p* = 0.039).

### 2.3. Genotype Combinations of miRNA Biogenesis Gene Polymorphisms

We next performed genotype combination analyses for the miRNA biogenesis gene polymorphisms tested in this study (Table 3). These data revealed that the *RAN* rs14035 CC/*XPO5* rs11077 AC genotype was significantly more frequent in VTE patients than in control subjects (AOR = 2.061, 95% CI = 1.153–3.686, FDR-adjusted *p* = 0.045). Similarly, the *DICER1* rs3742330 AA/*XPO5* rs11077 AC and *DROSHA* rs10719 TT/*XPO5* rs11077 AC genotypes were more frequently observed in VTE patients (AOR = 4.326, 95% CI = 2.024–9.245, FDR-adjusted *p* = 0.001; AOR = 2.385, 95% CI = 1.239–4.590, FDR-adjusted *p* = 0.047, respectively). In contrast, the frequencies of the *RAN* rs14035 CT/*XPO5* rs11077 AA (AOR = 0.583, 95% CI = 0.373–0.912, FDR-adjusted *p* = 0.045) and *DROSHA* rs10719 TT/*RAN* rs14035 CT (AOR = 0.445, 95% CI = 0.250–0.790, FDR-adjusted *p* = 0.040) genotypes were significantly higher in control participants.

We also performed genotype combination analysis in unprovoked VTE patients (Table 3) and found that the *DICER1* rs3742330 AA/*XPO5* rs11077 AC genotypes were more frequently observed in VTE patients (AOR = 4.709, 95% CI = 1.928–11.502, FDR-adjusted *p* = 0.004). In contrast, the *DICER1* rs3742330 AG/*RAN* rs14035 CT, *DROSHA* rs10719 TT/*RAN* rs14035 CT, and *RAN* rs14035 CT/*XPO5* AA rs11077 genotypes were more commonly observed in control participants (AOR = 0.153, 95% CI = 0.048–0.491, FDR-adjusted *p* = 0.010; AOR = 0.318, 95% CI = 0.138–0.731, FDR-adjusted *p* = 0.042; AOR = 0.378, 95% CI = 0.197–0.726, FDR-adjusted *p* = 0.014, respectively).

### 2.4. Genotype Combinations of miRNA Biogenesis Gene Polymorphisms

We then analyzed allele combinations of the four miRNA biogenesis genes polymorphisms in VTE patients and control subjects (Table 4). When total VTE patients were compared with controls, the C-C (*RAN*/*XPO5*), T-C-C (*DROSHA*/*RAN*/*XPO5*), A-T-C (*DICER1*/*RAN*/*XPO5*), A-T-C-C (*DICER1*/*DROSHA*/*RAN*/*XPO5*), and A-T-T-C (*DICER1*/*DROSHA*/*RAN*/*XPO5*) genotypes were significantly associated with an increased risk of VTE (*p* < 0.05). Conversely, the T-A (*RAN*/*XPO5*), T-T-A (*DROSHA*/*RAN*/*XPO5*), and A-T-T-A (*DICER1*/*DROSHA*/*RAN*/*XPO5*) genotypes were significantly less frequent in patients with VTE (*p* < 0.05). In patients with unprovoked VTE, the C-C (*RAN*/*XPO5*) and T-C-C (*DROSHA*/*RAN*/*XPO5*) genotypes were associated with an increased risk of VTE (*p* < 0.05). However, the T-A (*RAN*/*XPO5*), T-T-A (*DROSHA*/*RAN*/*XPO5*), A-T-T-A (*DICER1*/*DROSHA*/*RAN*/*XPO5*) genotypes were significantly less frequent in patients with VTE (*p* < 0.01).

## 3. Discussion

Hemostasis has the dual role of retaining blood flow and preventing inappropriate coagulation [11]. Previous studies have demonstrated that miRNAs and miRNA biogenesis genes (e.g., *DICER1* and *DROSHA*) play important roles in vascular biology and hemostasis. For example, the platelet miRNA profiles are associated with platelet functions, miRNA biogenesis gene may alter platelet miRNA expression. When *DICER1* gene, one of the miRNA biogenesis genes, was deleted, platelet miRNA profile was changed, and platelet reactivity was enhanced [37,38]. In particular, miRNA-145 is a significant factor in the prevention of thrombosis production and is present at decreased levels in VTE patients [14]. Despite the fact that the miRNA biogenesis gene haves been studied in other diseases [39,40,41]; however, the miRNA modulation mechanisms in VTE patients remain unclear. These miRNA biogenesis genes, including *DICER1*, *DROSHA*, *RAN*, and *XPO5*, play critical roles in miRNA production. As such, numerous other studies have found that miRNA biogenesis gene knockdown leads to a global reduction in mature miRNA levels. Those miRNAs for which decreased levels have been demonstrated are shown in Appendix A [23,24,25,26,27]. In this study, we analyzed the association between polymorphisms in each of the four miRNA biogenesis genes (*DICER1*, *DROSHA*, *RAN*, *XPO5*) and the risk of VTE in Korean subjects. From this analysis, we found that the *RAN* rs14035C > T and *XPO5* rs11077A > C polymorphisms are associated with total VTE and unprovoked VTE susceptibility. Additionally, in our Genotype combination and allele combination data, we found that *RAN* rs14035 T allele was associated with decreased risk of VTE and *XPO5* rs11077 C allele was associated with increased risk of VTE [36].

RAN is small Ras-related GTP-binding protein that plays an important role in the cell cycle. Specifically, RAN functions in the transport of molecules from the nucleus to the cytoplasm or from the cytoplasm to the nucleus through the nuclear pore complex, in a GTP-dependent manner [42]. In miRNA biology, RAN associates with a protein known as XPO5 to transport pre-miRNA precursors from the nucleus to the cytoplasm. Exportin (XPO) proteins are members of the karyopherin β family of transport factors, and XPO5 is one of the nucleo-cytoplasmic exportins. RAN, XPO5, and the pre-miRNA molecule form a complex, which functions to transport the pre-miRNA through the nuclear pore complex and into the cytoplasm. Additionally, there is a correlation between the expression of RAN and XPO5. In breast cancer, when RAN GTPase was overexpressed, XPO5 level was also significantly increased.

The *XPO5* gene has been found to play a role in carcinogenesis, as it was reported that certain cancers (e.g., non-small-cell lung cancer, esophageal squamous cell carcinoma, gastric cancer, hepatocellular carcinoma, and thyroid cancer) express levels of XPO5 that are distinct from those detected in normal cells [43,44,45,46,47,48]. In particular, a number of studies have reported that *XPO5* polymorphisms at the rs11077 locus affect disease development and patient survival in various cancers. For example, the rs11077AA genotype displayed a trend for high expression in ESCC tissues, and these high XPO5 expression levels were also associated with high survival rates among ESCC patients [45]. In addition, the *XPO5* rs11077 polymorphism is associated with XPO5 stability and miRNA expression levels. One study reported that when a polymorphism rs11077 is present in *XPO5*, precursor miRNAs are trapped in the nucleus, and processing efficiency decreases. This results in reduced miRNA levels, leading to diminished miRNA target gene inhibition [27]. In another case, using a Renilla luciferase 3′-UTR reporter assay, the rs11077 CC genotype was shown to promote reduced Renilla expression system, and knockdown of XPO5 expression lead to reduced miRNA levels [43]. In addition, XPO5 functions to regulate expression of DICER1, another miRNA biogenesis factor that cleaves the pre-miRNA with a hairpin structure to miRNA; miRNA duplex form in the cytoplasm. Notably, it was shown that decreased levels of XPO5, lead to decreased DICER1 expression [49]. Overall, these data suggest that SNPs in miRNA biogenesis genes affect the expression of mature miRNAs and consequently, may also influence miRNA-mediated regulation.

Our data further indicate that the *DROSHA* rs10719T > C polymorphism is associated with the incidence of provoked VTE. DROSHA has previously been shown to play a role in vascular smooth muscle cell survival and angiogenesis [50], and provoked VTE results from risk factors associated with pregnancy, surgery, and hormones. Therefore, these findings suggest that in provoked VTE, DROSHA is significantly associated with vascular inflammation and recovery.

There are a number of limitations to this case-control study. For example, both the VTE patient and control sample sizes were small and the study population was comprised only of Korean individuals. We will therefore need to validate these data in other ethnic groups. In addition, although we found a significant association between the *RAN* rs14035 and *XPO5* rs11077 polymorphisms and VTE, a proposed mechanism for the role of these polymorphisms in VTE prevalence is lacking. Thus, additional studies, potentially analyzing the effects of other miRNA biogenesis genes, will be needed to clarify the nature of the association between these polymorphisms and VTE.

## 4. Materials and Methods

### 4.1. Ethics Statement

All study protocols for this genetic analysis were reviewed and approved by The Institutional Review Board of CHA Bundang Medical Center in 10 January 2012 (IRB-number: 2005-002) and followed the recommendations of the Declaration of Helsinki (Fifth revision. 7, October, 2000). Written informed consent was obtained from all study participants prior to enrollment.

### 4.2. Study Population

A set of 203 patients with a recent (<6 months) objective diagnosis of DVT or PE, who visited the CHA Bundang Medical Center (Seongnam, Korea) between March 2006 and January 2011, were enrolled in the study. This VTE group included only patients with symptomatic VTE and excluded individuals with asymptomatic VTE. VTE was classified as provoked if the patient presented with at least one of the following risk factors: recent surgery (<3 months), recent trauma/fracture (<3 months), immobilization (>7 days), malignancy, stroke, severe medical disease, autoimmune disease, pregnancy, use of oral contraceptives, or known hypercoagulable disease. VTE was classified as unprovoked if these risk factors were absent. The 300 control group was chosen among patients visiting the CHA Bundang Health Promotion Center for periodic health examinations, who had no medical history of VTE. Vascular risk factors were assessed using medical records and laboratory data at the patient’s first hospital appointment. The data included in this study consisted of age, sex, hypertension, diabetes, lipidemia. Hypertension was diagnosed when a subject had a high baseline blood pressure (≥systolic blood pressure (SBP) 140 mmHg or diastolic blood pressure (DBP) 90 mmHg) or patient had been taken anti-hypertensive medication. Diabetes mellitus was diagnosed when subject’s high fasting plasma glucose level was higher than 126 mg/dL or had been taken anti-diabetic drugs. Smoking indicates current smoker at the time of the examination. Hyperlipidemia was defined as fasting serum total cholesterol of 240 mg/dL or a history of anti-hyperlipidemic agent treatment [51].

### 4.3. Genotyping

Peripheral blood samples were collected in blood collection tube and were ethylenediaminetetraacetic acid (EDTA) treated. Blood samples were centrifuged at 3000 rpm for 15 min, the buffy coat layer was collected, and leukocytes were separated. DNA was extracted from subject leukocytes using the G-DEX II Genomic DNA Extraction Kit (Intron Biotechnology, Seongnam, Korea), according to manufacturer instructions. We then assayed the four best-studied single nucleotide polymorphisms (SNPs) in the miRNA biogenesis genes, as determined by a literature search, which included the following 3′-UTR SNPs: *DICER1* rs3742330A > G, *DROSHA* rs10719T > C, *RAN* rs14035C > T, and *XPO5* rs11077A > C. These miRNA biogenesis gene polymorphisms were analyzed by the polymerase chain reaction-restriction fragment length polymorphism (PCR-RFLP) method, and the PCR conditions for these analyses are presented in Appendix A.

### 4.4. Statistical Analysis

Clinical characteristics of the study subjects were compared using the independent sample t-test. Associations among VTE prevalence and the four miRNA biogenesis genotypes and allele combination frequencies were estimated by analyzing the odds ratios (ORs) and 95% confidence intervals (CIs) with multivariate logistic regression and the Fisher’s exact test, respectively. Adjusted ORs (AORs) for the polymorphisms were determined using multiple logistic regression analysis based on gender, age, diabetes mellitus, hypertension, hyperlipidemia, and smoking status. The genotype distribution for each polymorphism was assessed for Hardy-Weinberg equilibrium deviations, and the genotype and allele frequency differences between groups were assessed using χ2 tests. *p*-values < 0.05 were considered to be statistically significant. HAPSTAT software, version 3.0 (University of North Carolina, Chapel Hill, NC, USA) (www.bios.unc.edu/Elin/hapstat/) was used to estimate haplotype frequencies for polymorphisms that were determined by multifactor dimensionality reduction analyses to have strong synergistic effects. Statistical analyses to measure the association between *DICER1*, *DROSHA*, *RAN*, and *XPO5* polymorphisms and VTE prevalence were performed using MedCalc, version 18.9 (MedCalc Software, Mariakerke, Belgium) and GraphPad Prism, version 4.0 (GraphPad, San Diego, CA, USA) software. The false-positive discovery rate (FDR) correction was used to adjust for multiple comparison tests, and associations with FDR-adjusted *p*-values < 0.05 were considered to be significant.

## 5. Conclusions

In this study, we have identified an association between susceptibility to VTE and polymorphisms in the miRNA biogenesis genes, *RAN* rs14035C > T and *XPO5* rs11077A > C. These findings may provide the basis for continued research efforts focusing on the roles of *RAN* and *XPO5* in hemostasis and thrombus development. We propose that the *RAN* rs14035 and *XPO5* rs11077 polymorphisms influence miRNA biosynthesis, and therefore, affect miRNA post-transcriptional regulation during hemostasis (i.e., platelet biogenesis, coagulation, and anticoagulation) and thrombus formation. However, the mechanisms underlying these relationships remain to be elucidated in future research.

## Figures and Tables

**Table 1 ijms-20-03771-t001:** Baseline characteristics of venous thromboembolism (VTE) patients and control subjects.

Characteristic	Controls (*n* = 300)	VTE Patients (*n* = 203)	*p*-value ^a^	Unprovoked VTE (*n* = 93)	*p*-value ^a^
Age (years, mean ± SD)	57.18 ± 9.96	56.07 ± 17.79	0.881	57.23 ± 17.76	0.447
Male (%)	138 (46.0)	103 (50.7)	0.297	54 (58.1)	**0.042**
Hypertension (%)	94 (31.3)	61 (30.0)	0.760	31 (33.3)	0.718
DM (%)	26 (8.7)	31 (15.3)	0.022	17 (18.3)	**0.010**
Lipidemia (%)	51 (17.0)	41 (20.2)	0.363	16 (17.2)	0.964
Smoking (%)	104 (34.7)	58 (28.6)	0.152	33 (35.5)	0.885

^a^*p*-values were calculated using Chi-square test for categorical data and Mann-Whitney test for continuous data. Significant *p*-values < 0.05 are shown in bold. Abbreviations: DM, diabetes mellitus; SD, standard deviation; VTE, venous thromboembolism.

**Table 2 ijms-20-03771-t002:** Genotype frequencies of *DICER1*, *DROSHA*, *RAN*, and *XPO5* polymorphisms in VTE patients and controls.

Genotype	Controls	Total VTE	AOR (95% CI) ^a^	*p*-value ^b^	Unprovoked VTE	AOR (95% CI) ^a^	*p*-value ^b^	Provoked VTE	AOR (95% CI) ^a^	*p*-value ^b^
	(*n* = 300)	(*n* = 203)			(*n* = 93)			(*n* = 110)		
***DICER1***										
**rs3742330A>G**										
AA	109 (36.3)	79 (38.9)	1.000 (reference)		42 (45.2)	1.000 (reference)			1.000 (reference)	
AG	137 (45.7)	92 (45.3)	0.953 (0.638–1.424)	0.815	35 (37.6)	0.685 (0.403–1.164)	0.162	37 (33.6)	1.247 (0.756–2.057)	0.387
GG	54 (18.0)	32 (15.8)	0.788 (0.460–1.350)	0.386	16 (17.2)	0.729 (0.371–1.432)	0.359	57 (51.8)	0.830 (0.413–1.669)	0.602
Dominant			0.919 (0.633–1.336)	0.659		0.708 (0.438–1.144)	0.158	16 (14.5)	1.130 (0.704–1.815)	0.612
Recessive			0.873 (0.536–1.423)	0.586		0.971 (0.519–1.815)	0.926		0.765 (0.411–1.425)	0.399
HWE *p*-value	0.342	0.547								
***DROSHA***										
**rs10719T > C**										
TT	164 (54.7)	116 (57.1)	1.000 (reference)		50 (53.8)	1.000 (reference)		66 (60.0)	1.000 (reference)	
TC	123 (41.0)	72 (35.5)	0.892 (0.608–1.309)	0.559	39 (41.9)	1.101 (0.673–1.799)	0.703	33 (30.0)	0.696 (0.426–1.136)	0.147
CC	13 (4.3)	15 (7.4)	1.893 (0.837–4.278)	0.125	4 (4.3)	1.250 (0.379–4.120)	0.714	11 (10.0)	**2.200 (0.891–5.428)**	**0.087**
Dominant			0.970 (0.672–1.400)	0.871		1.096 (0.680–1.766)	0.707		0.828 (0.525–1.306)	0.418
Recessive			1.897 (0.872–4.127)	0.106		1.115 (0.349–3.563)	0.854		**2.460 (1.048–5.774)**	**0.039**
HWE *p*-value	0.089	0.414								
***RAN***										
**rs14035C > T**										
CC	178 (59.3)	141 (69.5)	1.000 (reference)		71 (76.3)	1.000 (reference)		70 (63.6)	1.000 (reference)	
CT	113 (37.7)	58 (28.6)	**0.630 (0.425–0.935)**	**0.022**	21 (22.6)	**0.433 (0.248–0.756)**	**0.003**	37 (33.6)	0.833 (0.519–1.338)	0.450
TT	9 (3.0)	4 (2.0)	0.569 (0.169–1.919)	0.363	1 (1.1)	0.268 (0.032–2.217)	0.222	3 (2.7)	0.973 (0.249–3.796)	0.969
Dominant			**0.627 (0.427–0.922)**	**0.018**		**0.423 (0.245–0.730)**	**0.002**		0.842 (0.531–1.337)	0.467
Recessive			0.684 (0.205–2.275)	0.535		0.368 (0.045–2.981)	0.349		1.014 (0.264–3.899)	0.984
HWE *p*-value	0.073	0.482								
***XPO5***										
**rs11077A > C**										
AA	263 (87.7)	148 (72.9)	1.000 (reference)		67 (72.0)	1.000 (reference)		81 (73.6)	1.000 (reference)	
AC	36 (12.0)	54 (26.6)	**2.522 (1.564–4.067)**	**<0.001**	25 (26.9)	**2.611 (1.450–4.700)**	**0.001**	29 (26.4)	**2.398 (1.359–4.233)**	**0.003**
CC	1 (0.3)	1 (0.5)	1.542 (0.093–25.601)	0.763	1 (1.1)	2.971 (0.175–50.478)	0.451	0 (0.0)	-	-
Dominant			**2.493 (1.552–4.003)**	**<0.001**		**2.624 (1.469–4.689)**	**0.001**		**2.336 (1.326–4.116)**	**0.003**
Recessive			1.119 (0.068–18.398)	0.937		2.516 (0.150-42.109)	0.521		-	-
HWE *p*-value	0.843	0.091								

^a^ Adjusted by age, gender, hypertension, diabetes mellitus, lipidemia, and smoking status. ^b^
*p*-values were calculated using logistic regression analysis. Significant *p*-values < 0.05 are shown in bold. Abbreviations: AOR, adjusted odds ratio; 95% CI, 95% confidence interval; HWE, Hardy-Weinberg equilibrium; VTE, venous thromboembolism.

**Table 3 ijms-20-03771-t003:** Genotype combination analysis for *DICER1*, *DROSHA*, *RAN*, and *XPO5* polymorphisms in VTE patients and control subjects.

Genotype	Controls (*n* = 300)	Total VTE (*n* = 203)	AOR (95% CI) ^a^	*p*-value ^b^	FDR-Adjusted *p*-value	Unprovoked VTE (*n* = 93)	AOR (95% CI) ^a^	*p*-value ^b^	FDR-Adjusted *p*-value
***DICER1/RAN***									
AA-CC	60 (20.0)	51 (25.1)	1.000 (reference)			29 (31.2)	1.000 (reference)		
AA-CT	46 (15.3)	26 (12.8)	0.658 (0.349–1.240)	0.195	0.520	12 (12.9)	0.446 (0.192–1.038)	0.061	0.183
AA-TT	3 (1.0)	2 (1.0)	0.690 (0.099–4.812)	0.708	0.796	1 (1.1)	0.627 (0.056–7.036)	0.705	0.705
AG-CC	88 (29.3)	68 (33.5)	0.936 (0.567–1.545)	0.796	0.796	31 (33.3)	0.725 (0.386–1.359)	0.315	0.473
GG-CC	30 (10.0)	22 (10.8)	0.832 (0.417–1.659)	0.602	0.796	11 (11.8)	0.729 (0.312–1.703)	0.465	0.558
AG-CT	45 (15.0)	23 (11.3)	0.569 (0.298–1.086)	0.088	0.350	4 (4.3)	**0.153 (0.048–0.491)**	**0.002**	**0.010**
AG-TT	4 (1.3)	1 (0.5)	0.390 (0.040–3.798)	0.417	0.796	0 (0.0)	N/A	N/A	N/A
GG-CT	22 (7.3)	9 (4.4)	0.433 (0.178–1.054)	0.065	0.350	5 (5.4)	0.413 (0.137–1.246)	0.117	0.233
GG-TT	2 (0.7)	1 (0.5)	0.610 (0.045–8.209)	0.710	0.796	0 (0.0)	N/A	N/A	N/A
***DICER1/XPO5***									
AA-AA	95 (31.7)	49 (24.1)	1.000 (reference)			26 (28.0)	1.000 (reference)		
AA-AC	14 (4.7)	30 (14.8)	**4.326 (2.024–9.245)**	**0.0002**	**0.001**	16 (17.2)	**4.709 (1.928–11.502)**	**0.001**	**0.004**
AA-CC	0 (0.0)	0 (0.0)	N/A	N/A	N/A	0 (0.0)	N/A	N/A	N/A
AG-AA	120 (40.0)	72 (35.5)	1.233 (0.772–1.968)	0.381	0.634	29 (31.2)	1.008 (0.541–1.879)	0.979	0.979
GG-AA	48 (16.0)	27 (13.3)	1.017 (0.555–1.862)	0.958	0.958	12 (12.9)	0.867 (0.392–1.920)	0.725	0.979
AG-AC	16 (5.3)	20 (9.9)	**2.387 (1.112–5.125)**	**0.026**	0.064	6 (6.5)	1.160 (0.382–3.519)	0.793	0.979
AG-CC	1 (0.3)	0 (0.0)	N/A	N/A	N/A	0 (0.0)	N/A	N/A	N/A
GG-AC	6 (2.0)	4 (2.0)	1.358 (0.352–5.236)	0.657	0.821	3 (3.2)	1.994 (0.436–9.116)	0.374	0.934
GG-CC	0 (0.0)	1 (0.5)	N/A	N/A	N/A	1 (1.1)	N/A	N/A	N/A
***DROSHA/RAN***									
TT-CC	96 (32.0)	86 (42.4)	1.000 (reference)			40 (43.0)	1.000 (reference)		
TT-CT	61 (20.3)	27 (13.3)	**0.445 (0.250–0.790)**	**0.006**	**0.040**	10 (10.8)	**0.318 (0.138–0.731)**	**0.007**	**0.042**
TT-TT	7 (2.3)	3 (1.5)	0.663 (0.162–2.709)	0.567	0.661	0 (0.0)	N/A	N/A	N/A
TC-CC	75 (25.0)	45 (22.2)	0.692 (0.427–1.121)	0.135	0.436	28 (30.1)	0.889 (0.494–1.602)	0.696	0.914
CC-CC	7 (2.3)	10 (4.9)	1.858 (0.648–5.325)	0.249	0.436	3 (3.2)	1.213 (0.283–5.205)	0.795	0.914
TC-CT	46 (15.3)	26 (12.8)	0.684 (0.383–1.221)	0.199	0.436	10 (10.8)	0.496 (0.219–1.124)	0.093	0.279
TC-TT	2 (0.7)	1 (0.5)	0.373 (0.030–4.595)	0.442	0.618	1 (1.1)	1.152 (0.090–14.704)	0.914	0.914
CC-CT	6 (2.0)	5 (2.5)	1.084 (0.308–3.818)	0.901	0.901	1 (1.1)	0.459 (0.051–4.143)	0.488	0.914
CC-TT	0 (0.0)	0 (0.0)	N/A	N/A	N/A	0 (0.0)	N/A	N/A	N/A
***DROSHA/XPO5***									
TT-AA	143 (47.7)	84 (41.4)	1.000 (reference)			36 (38.7)	1.000 (reference)		
TT-AC	21 (7.0)	31 (15.3)	**2.385 (1.239–4.590)**	**0.009**	**0.047**	13 (14.0)	**2.304 (1.017–5.217)**	**0.045**	0.136
TT-CC	0 (0.0)	1 (0.5)	N/A	N/A	N/A	1 (1.1)	N/A	N/A	N/A
TC-AA	109 (36.3)	54 (26.6)	0.913 (0.591–1.410)	0.680	0.680	28 (30.1)	1.119 (0.631–1.985)	0.700	0.757
CC-AA	11 (3.7)	10 (4.9)	1.776 (0.700–4.504)	0.226	0.283	3 (3.2)	1.242 (0.315–4.894)	0.757	0.757
TC-AC	13 (4.3)	18 (8.9)	**2.624 (1.156–5.955)**	**0.021**	0.053	11 (11.8)	**3.346 (1.276–8.774)**	**0.014**	0.084
TC-CC	1 (0.3)	0 (0.0)	N/A	N/A	N/A	0 (0.0)	N/A	N/A	N/A
CC-AC	2 (0.7)	5 (2.5)	5.425 (0.948–31.053)	0.058	0.096	1 (1.1)	3.820 (0.292–49.938)	0.307	0.614
CC-CC	0 (0.0)	0 (0.0)	N/A	N/A	N/A	0 (0.0)	N/A	N/A	N/A
***RAN/XPO5***									
CC-AA	153 (51.0)	103 (50.7)	1.000 (reference)			52 (55.9)	1.000 (reference)		
CC-AC	25 (8.3)	37 (18.2)	**2.061 (1.153–3.686)**	**0.015**	**0.045**	18 (19.4)	1.965 (0.971–3.975)	0.060	0.120
CC-CC	0 (0.0)	1 (0.5)	N /A	N/A	N/A	1 (1.1)	N/A	N/A	N/A
CT-AA	103 (34.3)	43 (21.2)	**0.583 (0.373–0.912)**	**0.018**	**0.045**	15 (16.1)	**0.378 (0.197–0.726)**	**0.004**	**0.014**
TT-AA	7 (2.3)	2 (1.0)	0.431 (0.085–2.179)	0.309	0.386	0 (0.0)	N/A	N/A	N/A
CT-AC	9 (3.0)	15 (7.4)	2.191 (0.898–5.346)	0.085	0.142	6 (6.5)	1.368 (0.432–4.338)	0.594	0.698
CT-CC	1 (0.3)	0 (0.0)	N/A	N/A	N/A	0 (0.0)	N/A	N/A	N/A
TT-AC	2 (0.7)	2 (1.0)	1.507 (0.201–11.300)	0.690	0.690	1 (1.1)	1.638 (0.135–19.801)	0.698	0.698
TT-CC	0 (0.0)	0 (0.0)	N/A	N/A	N/A	0 (0.0)	N/A	N/A	N/A

^a^ Calculated on the basis of risk factors, including age, gender, HTN, DM, lipidemia, and smoking status. ^b^
*P*-values were calculated using Chi-square test and Fisher’s exact test; significant *p*-values < 0.05 are shown in bold. Abbreviations: AOR, adjusted odds ratio; 95% CI, 95% confidence interval; FDR, false-positive discovery rate; VTE, venous thromboembolism.

**Table 4 ijms-20-03771-t004:** Allele combination analysis of the *DICER1*, *DROSHA*, *RAN*, and *XPO5* gene polymorphisms in VTE patients and controls.

Allele Combination	Controls (2*n* = 600)	Total VTE (2*n* = 406)	OR (95% CI)	*p*-value ^a^	FDR-Adjusted *p*-value	Unprovoked VTE (2*n* = 186)	OR (95% CI)	*p*-value ^a^	FDR-Adjusted *p*-value
***DICER1*/*DROSHA*/*RAN*/*XPO5***									
A-T-C-A	184 (30.6)	137 (33.7)	1.000 (reference)			74 (39.9)	1.000 (reference)		
A-T-C-C	13 (2.1)	21 (5.2)	**2.170 (1.049–4.486)**	**0.033**	0.144	11 (6.0)	2.104 (0.902–4.909)	0.080	0.173
A-T-T-A	64 (10.7)	16 (4.0)	**0.336 (0.186–0.606)**	**0.0002**	**0.003**	3 (1.8)	**0.117 (0.035–0.383)**	**<0.0001**	**0.001**
A-T-T-C	2 (0.4)	10 (2.4)	**6.715 (1.447–31.160)**	**0.005**	**0.035**	4 (2.4)	4.973 (0.891–27.750)	0.065	0.168
A-C-C-A	71 (11.9)	34 (8.4)	0.643 (0.404–1.024)	0.062	0.181	13 (7.0)	**0.455 (0.238–0.872)**	**0.016**	0.051
A-C-C-C	9 (1.5)	13 (3.2)	1.940 (0.806–4.670)	0.182	0.295	4 (2.4)	1.105 (0.330–3.701)	1.000	1.000
A-C-T-A	13 (2.1)	19 (4.7)	1.963 (0.937–4.112)	0.070	0.181	8 (4.5)	1.530 (0.609–3.845)	0.363	0.524
A-C-T-C	0 (0)	0 (0.0)	N/A	N/A	N/A	0 (0.0)	N/A	N/A	N/A
G-T-C-A	142 (23.7)	91 (22.3)	0.861 (0.610–1.214)	0.392	0.464	32 (17.3)	**0.560 (0.351–0.896)**	**0.015**	0.051
G-T-C-C	10 (1.6)	12 (2.9)	1.612 (0.677–3.840)	0.277	0.361	7 (3.7)	1.741 (0.638–4.746)	0.274	0.445
G-T-T-A	34 (5.6)	18 (4.4)	0.711 (0.385–1.312)	0.274	0.188	7 (3.7)	0.512 (0.217–1.206)	0.120	0.051
G-T-T-C	4 (0.6)	0 (0.0)	0.149 (0.008–2.794)	0.141	0.188	0 (0.0)	0.275 (0.015–5.178)	0.580	0.051
G-C-C-A	41 (6.8)	32 (8.0)	1.048 (0.628–1.750)	0.857	0.188	21 (11.3)	1.274 (0.705–2.300)	0.422	0.051
G-C-T-A	14 (2.3)	4 (0.9)	0.384 (0.124–1.192)	0.087	0.188	0 (0.0)	**0.085 (0.005–1.451)**	**0.014**	0.051
G-C-T-C	1 (0.1)	0 (0.0)	0.447 (0.018–11.070)	1.000	0.188	0 (0.0)	0.826 (0.033–20.510)	1.000	0.051
***DICER1*/*RAN*/*XPO5***									
A-C-A	254 (42.3)	175 (43.0)	1.000 (reference)			89 (47.6)	1.000 (reference)		
A-C-C	22 (3.6)	33 (8.1)	**2.177 (1.228–3.861)**	**0.007**	**0.023**	15 (7.9)	1.946 (0.967–3.916)	0.059	0.103
A-T-A	77 (12.8)	32 (7.8)	**0.603 (0.383–0.951)**	**0.028**	0.066	11 (6.1)	**0.408 (0.207–0.802)**	**0.008**	0.054
A-T-C	2 (0.4)	11 (2.7)	**7.983 (1.747–36.470)**	**0.002**	**0.011**	4 (2.4)	**5.708 (1.027–31.710)**	**0.046**	0.103
G-C-A	184 (30.7)	120 (29.6)	0.947 (0.701–1.278)	0.720	0.720	52 (28.0)	0.807 (0.545–1.193)	0.281	0.281
G-C-C	9 (1.5)	12 (3.0)	1.935 (0.798–4.692)	0.138	0.193	8 (4.2)	2.537 (0.950–6.778)	0.088	0.123
G-T-A	47 (7.9)	24 (5.8)	0.741 (0.437–1.257)	0.265	0.309	7 (3.8)	**0.425 (0.185–0.975)**	**0.038**	0.103
G-T-C	5 (0.8)	0 (0.0)	0.132 (0.007–2.401)	0.085	0.148	0 (0.0)	0.259 (0.014–4.725)	0.187	0.218
***DROSHA*/RAN/XPO5**									
T-C-A	326 (54.3)	231 (56.8)	1.000 (reference)			108 (58.3)	1.000 (reference)		
T-C-C	23 (3.8)	32 (7.8)	**1.963 (1.120–3.444)**	**0.017**	0.056	16 (8.8)	**2.100 (1.070–4.121)**	**0.028**	0.098
T-T-A	97 (16.1)	32 (7.9)	**0.466 (0.302–0.718)**	**0.001**	**0.004**	9 (4.8)	**0.280 (0.137–0.574)**	**0.0002**	**0.001**
T-T-C	6 (1)	10 (2.5)	2.352 (0.843–6.564)	0.093	0.163	5 (2.7)	2.515 (0.752–8.409)	0.157	0.274
C-C-A	112 (18.7)	63 (15.6)	0.794 (0.558–1.129)	0.198	0.277	33 (17.5)	0.889 (0.570–1.388)	0.606	0.848
C-C-C	8 (1.4)	15 (3.6)	**2.646 (1.103–6.346)**	**0.024**	0.056	6 (3.0)	2.264 (0.768–6.672)	0.129	0.274
C-T-A	28 (4.6)	24 (5.9)	1.210 (0.684–2.141)	0.513	0.598	9 (4.8)	0.970 (0.444–2.121)	0.940	1.000
C-T-C	1 (0.2)	0 (0.0)	0.470 (0.019–11.600)	1.000	1.000	0 (0.0)	1.003 (0.041–24.820)	1.000	1.000
***RAN*/*XPO5***									
C-A	438 (73)	294 (72.4)	1.000 (reference)			141 (75.7)	1.000 (reference)		
C-C	31 (5.2)	46 (11.3)	**2.211 (1.369–3.569)**	**0.001**	**0.003**	22 (12.0)	**2.205 (1.236–3.932)**	**0.006**	**0.009**
T-A	124 (20.7)	56 (13.8)	**0.673 (0.475–0.953)**	**0.025**	**0.038**	18 (9.8)	**0.451 (0.266–0.766)**	**0.003**	**0.008**
T-C	7 (1.2)	10 (2.5)	2.128 (0.801–5.656)	0.121	0.121	5 (2.5)	2.219 (0.693–7.103)	0.181	0.181

^a^*P*-values were calculated using Chi-square test and Fisher’s exact test; significant *p*-values < 0.05 are shown in bold. Abbreviations: OR, odds ratio; CI, confidence interval; FDR, false-positive discovery rate; OD, odds ratio; VTE, venous thromboembolism.

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
