# Peer review of "Analysis of the Association Between MicroRNA Biogenesis Gene Polymorphisms and Venous Thromboembolism in Koreans"

_ijms, 2019, doi:10.3390/ijms20153771_

Round 1

Reviewer 1 Report

The quality of the research is high. But the problem is limited to the Korean population. This does not allow for large-scale parallels. But this is a great precedent for research in other populations. This allows you to rate the article highly.

Author Response

Reviewer 1

Comments and Suggestions for Authors

The quality of the research is high. But the problem is limited to the Korean population. This does not allow for large-scale parallels. But this is a great precedent for research in other populations. This allows you to rate the article highly.

=> Thank you for critical comments. We described this limitation in the discussion section, page 12, line 231-233.

Reviewer 2 Report

The authors analyzed four SNPs in the 3’UTR of four miRNA biogenesis related genes: DICER, DROSHA, RAN and XPO5. These polymorphic loci were then been analyzed in relation with VTE in the group of patients and control individual from Korea. The SNP analysis was done by PCR-RFLP method. The number of patients from groups of provoke and unprovoked VTE are limited, but the authors mentioned this limitation in the discussion. Presented results are sound and convincing, however, nowadays when many large scale association studies  are published frequently, the authors results may be of interest for moderate group of scientists and clinicians working with VTE. In my opinion this manuscript provide novel results but there are some minor issues:

-          Line58: – The authors state that miRNA are transcribed by RNAPIII. What is the source of this information? To my best knowledge miRNA are mainly transcribed by RNAPII, and only some (for example of viral origin) are transcribed by RNAPIII

-           Lines62-63: - …, they are processed by the DICER1/TRBP complex to form mature, duplex  miRNA.. – usually the term “mature miRNA” is used for single stranded miRNA, after the digestion of passenger strain. It should be corrected.

-          Line120: “In contrast, the RAN rs14035C>T polymorphism was associated with reduced VTE risk”. How authors determined reduced VTE risk ? Is it only the higher frequency of this variant in control group or something else ? It should be specified.

-          The authors should follow the actual gene name and symbols nomenclature. For example the official symbol for Dicer in NCBI is DICER1 – italicized each time when the gene or RNA is meant.  It should be corrected within the whole manuscript for all genes.

-          Line 203-204 – In my opinion citation is missing.

In summary, this manuscript is worth to be published (but with moderate priority) after minor revision.

Author Response

Reviewer 2

Comments and Suggestions for Authors

The authors analyzed four SNPs in the 3’UTR of four miRNA biogenesis related genes: DICER, DROSHA, RAN and XPO5. These polymorphic loci were then been analyzed in relation with VTE in the group of patients and control individual from Korea. The SNP analysis was done by PCR-RFLP method. The number of patients from groups of provoke and unprovoked VTE are limited, but the authors mentioned this limitation in the discussion. Presented results are sound and convincing, however, nowadays when many large scale association studies are published frequently, the authors results may be of interest for moderate group of scientists and clinicians working with VTE. In my opinion this manuscript provide novel results but there are some minor issues:

=> Thank you for critical comments. We tried to augment some contents as you suggested. We revised our insufficient descriptions as follows:

-          Line58: – The authors state that miRNA are transcribed by RNAPIII. What is the source of this information? To my best knowledge miRNA are mainly transcribed by RNAPII, and only some (for example of viral origin) are transcribed by RNAPIII

=> Thank you for comments. Sorry for confusing to you. We made a typing error and we revised misspell.

“. The miRNA-encoding genes are transcribed by RNA polymerase Ⅱ as long, primary miRNAs (pri-miRNA) with a stem-loop structure (100-1000 nts), that are cleaved by the DROSHA/DGCR8 complex in nucleus.” [Introduction section, page 2, line 60-62]

-           Lines62-63: - …, they are processed by the DICER1/TRBP complex to form mature, duplex  miRNA.. – usually the term “mature miRNA” is used for single stranded miRNA, after the digestion of passenger strain. It should be corrected.

=> Thank you for comments. Your comment is appropriate and we agreed. We revised sentence.

“There, they are processed by the DICER1/TRBP complex to form miRNA duplex. One strand of the miRNA duplex is loaded onto Argonaute (AGO) to form the RNA-induced silencing complex (RISC), which binds to the 3’-untranslated region (UTR) of an mRNA transcript and functions to suppress translation or promote mRNA degradation, through a mechanism known as post-transcriptional gene silencing (PTGS)” [Introduction section, page 2 , line 64-68]

-          Line120: “In contrast, the RAN rs14035C>T polymorphism was associated with reduced VTE risk”. How authors determined reduced VTE risk ? Is it only the higher frequency of this variant in control group or something else ? It should be specified.

=> Thank you for comments. We revised and added the following paragraph to the result section

“In contrast, the RAN CT genotype was more frequent in control group than in VTE patients, and the RAN rs14035C>T polymorphism was associated with reduced VTE risk (CC vs. CT: AOR=0.630, 95% CI=0.425-0.935, P=0.022; CC vs. CT+TT: AOR=0.627, 95% CI=0.427-0.922, P=0.018)” [Result section, page 3, line 123-126]  

-          The authors should follow the actual gene name and symbols nomenclature. For example the official symbol for Dicer in NCBI is DICER1 – italicized each time when the gene or RNA is meant.  It should be corrected within the whole manuscript for all genes.

=> Thank you for comments. Your comment is appropriate and we agree. We changed the word to’ DICER1’ from 'DICER’ and italicized the gene name.

-          Line 203-204 – In my opinion citation is missing.

=> Thank you for comments. Sorry for confusing you. We revised sentence.

“For example, the rs11077 AA genotype displayed a trend for high expression in ESCC tissues, and these high XPO5 expression levels were also associated with high survival rates among ESCC patients.” [Discussion section, page 11, line 213-215]

In summary, this manuscript is worth to be published (but with moderate priority) after minor revision.
